# A Two-Stage Method for Online Signature Verification Using Shape Contexts and Function Features [note 1]

**DOI:** 10.3390/s19081808

**Published:** 2019-04-16

**Authors:** Yu Jia, Linlin Huang, Houjin Chen

**Affiliations:** School of Electronic and Information Engineering, Beijing Jiaotong University, No. 3 Shangyuancun Haidian District, Beijing 100044, China; 16120010@bjtu.edu.cn (Y.J.); hjchen@bjtu.edu.cn (H.C.)

**Keywords:** online signature verification, shape contexts, function features, SC-DTW, symbolic representation, two-stage method

## Abstract

As a behavioral biometric trait, an online signature is extensively used to verify a person’s identity in many applications. In this paper, we present a method using shape contexts and function features as well as a two-stage strategy for accurate online signature verification. Specifically, in the first stage, features of shape contexts are extracted from the input and classification is made based on distance metric. Only the inputs passing by the first stage are represented by a set of function features and verified. To improve the matching accuracy and efficiency, we propose shape context-dynamic time warping (SC-DTW) to compare the test signature with the enrolled reference ones based on the extracted function features. Then, classification based on interval-valued symbolic representation is employed to decide if the test signature is a genuine one. The proposed method is evaluated on SVC2004 Task 2 achieving an Equal Error Rate of 2.39% which is competitive to the state-of-the-art approaches. The experiment results demonstrate the effectiveness of the proposed method.

## 1. Introduction

Biometric verification technology has aroused a lot of interest due to its reliability, effectiveness, and convenience in verifying personal identity [1]. Verification techniques based on face [2], fingerprint, and some such physiological biometric attributes have brought extra convenience and changed our lifestyle [3]. Although behavioral biometric attributes are slightly inferior to physiological ones in stability and uniqueness, they are more accessible and less intrusive to users. Voice, signature, gait, etc. are all typical behavioral attributes. Among them, signature remains the most widespread and recognized socially and legally verification approach in our day-to-day life [4]. Signing is a customary and fast movement driven by long-term nervous system and writing habit. Therefore, signature verification techniques can have more potential applications in the real world.

Depending on the different methods of signature acquisition, signature verification technique can be split into two categories: offline and online. In the system of offline signature verification [5], images containing signatures are collected after finishing the signing process. For online signature verification, signatures are captured by sensor-based devices while the user is signing and represented by a set of temporal functions, from which both static and dynamic features are extracted and then used to make a decision on whether the signature belongs to its claimed user. Compared with offline signature verification, the dynamic information collection of online signature ensures its uniqueness and higher difficulty to forge, so online signature verification technique usually owns better performance in accuracy and security.

There are two parts of an online signature verification system: enrolment and verification. Several signatures are provided as reference signatures by the users during enrolment and their extracted features along with calculated thresholds would be stored in the knowledge base. In verification, the authenticity of a test signature is evaluated by matching its features with those from reference signatures of its claimed user [6].

Online signatures are collected by electronic devices such as tablets, smart phones, and so on. Most of them use sensors to capture various real-time data such as coordinates, pressure, timestamp, etc. during signing. After collection, the signatures are represented as time series and then undergo preprocessing and feature extraction modules successively.

Online signature verification methods can be categorized based on the feature extraction process and matching strategy [7]. According to the employed features, there are broadly two groups: parametric and function features-based approaches. In the framework of parametric features-based methods, a signature is characterized as a vector of elements and each one is a representative of the value of one feature [8]. Examples of such attributes are width, height, average speed, etc. The dimensions of parametric features of signatures are all equal. In the function features-based method, a signature is represented by a multi-dimension feature set constituted by several time functions. Coordinate, timestamp, pressure, etc. are commonly used function features. Generally, the function features-based approaches perform better due to more dynamic information application, but these kinds of method consume more computational time and memory.

With regards to the matching methods, distance-based and model-based approaches are two main techniques [9]. Dynamic time warping (DTW) has been often adopted in distance-based methods [10]. DTW is a well-known approach for aligning vectors of different lengths. For application in signature verification, a set of features at each sample point is extracted and the similarity between the test signatures and enrolled reference signatures is then computed using dynamic programming. Point-based warping technique is a variant of DTW, wherein only selective points are warped. Extreme point [11] and stroke point [12] are often used. In addition, some works make a fusion of DTW with other methods. Sharma and Sundaram [9] propose a method that uses the information from DTW cost matrix and warping paths alignments. The decision is made by the conjunction of warping path score and DTW score. Yanikoglu and Kholmatov [13] fuse the Fast Fourier Transform with DTW and the fusion system lowers the error rate by up to about 25%. Chen and Xia [14] extract a set of function features for comparing the dissimilarity-based DTW between the test signature and the template database. In addition, the nearest template and majority vote are proposed to classify. Model-based approaches employ either generative-based classifiers such as hidden Markov model (HMM) [15,16,17] or discriminative ones such as neural network (NN) [18,19,20] and support vector machine (SVM) [21,22]. Also, there are some hybrid methods that combine different methods mentioned above. Multi-stage cascade framework [23], multi-stage decision-level score fusion [24,25] or a multi-expert system for signature verification [26,27] have been reported in the literature. Recently, inspired by the great success of recurrent neural networks (RNNs) in sequential modeling, several verification methods based on RNNs are proposed. Lai et al. [28] propose a novel descriptor called the length-normalized path signature (LNPS) for feature representation and then features are fed into the GRU (Gated recurrent unit) network. Triplet loss and center loss were used to train the network with the BP algorithm. The method proposed in [29] extracts 23 hand-crafted time function features and uses the bidirectional LSTM (Long short-term memory) and GRU networks with Siamese architecture to learn a dissimilarity metric from the pairs of signatures.

Although it is not that easy for a forger to fake a signature that is exactly the same as the genuine one, due to the large intra-class variations from one person and small inter-class variations between forgeries and genuine ones, accurate online signature verification still remains a challenging problem.

In real applications, the forgeries are usually classified to be two types, named skilled forgery and random one. A skilled forgery is signed by a person who had access to the genuine signatures and practiced for a while. A random forgery is signed without with any information about the signature, or even the name of the person whose signature is forged [30]. Compared with skilled forgeries, the random forgeries are more common in our daily life. Obviously, the skilled forgeries are more difficult to verify. In addition, the loss brought by accepting forgeries is higher than that by rejecting genuine signatures, which means accepting a signature as genuine should be stricter. Considering these factors, we propose a two-stage method using shape contexts and function features for accurate online signature verification. Features of shape contexts are extracted from the input firstly and classification of this stage is based on shape distance metric. Only the inputs passing by the first stage are represented by a set of function features and verified. To improve the matching accuracy and efficiency, we employ a shape context-dynamic time warping (SC-DTW) to compare the test signature with the enrolled reference ones based on the extracted function features. An interval-valued symbolic representation-based classifier is proposed to decide if the test signature is a genuine one.

The contributions of this paper are as follows:Based on the fact of unbalanced occurrence probability of skilled signature forgeries and random ones, a fast and accurate two-stage verification method is proposed.Shape context feature extractor is designed to describe global shape characteristics of signature for fast classification of random forgeries.SC-DTW is applied to fulfill comparison task and interval-valued-based representation classifier is proposed for final decision-making to achieve state-of-the-art verification performance.

This paper is an extended version of the one published in proceedings of PRCV2018 [31]. In this paper, more details on feature extraction and matching methods are given. Moreover, to further improve the performance method in the paper of PRCV2018, more effective features are extracted. Instead of distance metric classification, an interval-valued symbolic representation-based classifier is employed to enhance classification ability. Besides, more detailed experimental results are reported.

The rest of this paper is organized as follows. Section 2 details the methodology we proposed. Signature preprocessing is presented in Section 2.1. Section 2.2 presents the shape context descriptor and online signature verification method based on it. The function features extraction, feature alignment, and symbolic classifier are showed in Section 2.3. Section 2.4 discusses the two-stage verification protocol. The database used in our experiment, experimental results, and performance analysis are provided in Section 3. The conclusion is offered finally in Section 4.

## 2. Methodology

The diagram of the proposed method is shown in Figure 1. The input signature is first preprocessed for smoothing and normalization, and then it is fed into the shape context-based verification module, which does well in quickly distinguishing the random forgeries owing to their manifest differences in shape. Most obvious forgeries can be rule out in this stage. The signature passed through the first module is verified by function features-based verification module. This module achieves more accurate verification results due to the application of details in signature and decision fusion by interval-valued symbolic representation-based classifier.

### 2.1. Preprocessing

Captured by electronic devices, the time series of a signature are mixed with noises and fluctuations unavoidably. In addition, the acquired signatures of one individual vary with time or places, with the result that there are differences in size and location between signatures. Therefore, we firstly let the acquired signatures pass the preprocessing module to address those issues. The preprocessing module includes smoothing and normalization. Gaussian smoothing is employed to filter the artifacts and smooth the data. Then we adopt moment normalization technique [32] to standardize the size and location of acquired signatures.

Set the signature as $S=(s_{1},s_{2},\ldots ,s_{i},\ldots ,s_{N})$, $s_{i}=\left(x_{i},y_{i}\right)$. N is the number of sample points, $\left(x_{i},y_{i}\right)$ is x and y coordinates information.

In the moment normalization technique, the size of a signature is not the difference between maximum and minimum in horizontal and vertical directions, but the width and height of the window derived from its moment, as is show in Figure 2. Denote the width and height of window as W and H, given by
(1)$$W=4\sqrt{\mu_{20}},H=4\sqrt{\mu_{02}}$$
$\mu_{pq}$ denotes the center moment, and $\left(x_{c},y_{c}\right)$ denotes the signature’s centroid.
(2)$$\mu_{pq}=\sum_{x}\sum_{y}{\left(x-x_{c}\right)}^{p}{\left(y-y_{c}\right)}^{q}$$

After window calculation, the size normalization technique is implemented as follows. The heights of the signatures are normalized to a predetermined value that in this paper is 300. Moreover, the aspect ratio of before and after preprocessing remains consistent to keep the signature shape unchanged.
(3)$$\begin{array}{c} x^{\prime }=\alpha \times \left(x-x_{{}_{c}}\right)+x_{c}^{\prime }\\ y^{\prime }=\beta \times \left(y-y_{c}\right)+y_{c}^{\prime } \end{array}$$
where *x* and *y* are smoothed originate coordinates. $x^{\prime }$ and $y^{\prime }$ are normalized coordinates. $x_{c}^{\prime }$ and $y_{c}^{\prime }$ are the centroid of normalized signature. α and β are the ratio of the normalized signature size to its original size, given by
(4)$$\begin{array}{c} \alpha =\frac{W_{norm}}{W},\\ \beta =\frac{H_{norm}}{H},\\ \frac{W_{norm}}{H_{norm}}=\frac{W}{H} \end{array}$$
where $W_{norm}$ and $H_{norm}$ denote the normalized width and height.

Signatures are centered at (0, 0) to normalize their locations. After preprocessing, the signatures have the same size and location. In this paper, we did not employ translation normalization since we believe signature’s angle is an out-of-habit feature. Figure 2 shows some examples of original signatures and corresponding preprocessed signatures.

### 2.2. Shape Context-Based Online Signature Verification

In the methods proposed for online signature verification, the dynamics properties of the signatures, for example, velocity, pressure, acceleration, etc. are widely applied. However, the shape of signature contains very useful details, which is critical for distinguish a signature between forgery and genuine one. The method proposed by Gupta and Joyce [33] extracts the dynamics properties of position extreme points of signatures and achieved better performance. Features based on shape also have been successively applied in offline signature verification [34].

In this paper, we propose a verification method based on shape context features. Specifically, shape context descriptor [34,35] is used to extract features of a signature and a cost matrix is computed. After finding the best one-to-one matching between two signatures’ shape and modeling transformation, the measurable shape distance is used for classification. To further improve the efficiency, only trend-transition-points (TTPs) that can represent the shape of a signature roughly are used for calculating distance.

#### 2.2.1. Shape Context Feature Extraction

Shape context descriptor captures the distribution over relative positions of shape points and the connectivity properties between features points along curves. Therefore, shape context features not only provide global characterization of shape but also contain more contextual information within a certain range of a signature. Besides, shape context descriptor is designed in a way of describing shapes that allows for measuring shape similarity and the recovering of point correspondences. Traditionally, the first step is to randomly select a set of points that lie on the edges of two shapes separately. Here the shape of an online signature is represented by a set of sampled points which in this work is $\left(x_{i},y_{i}\right)$, i=1,2,…,N. *N* is the number of sampled points.

Figure 3 shows the shapes and shape context histograms of a reference signature, a genuine signature and a skilled forgery of one user. Because the writing speed is a kind of relatively fixed and unique information, the number and distribution of sample points between genuine signature and reference signature are more similar. Taking one point as the origin of polar coordinate, the shape contexts of this point can be represented using log-polar histogram. We set five bins for logr and 12 bins for θ. The number of neighboring points that fall into the very bin is just the histogram value. Figure 3d–f present the corresponding histograms for certain points. We can see that the difference in shape context histograms between genuine signature and reference signature is relatively small, while the histogram of skilled forgeries is quite dissimilar to the reference ones.

Considering a point $p_{i}$ on the first shape and a point $q_{j}$ on the second shape, denote $C_{ij}=C\left(p_{i},q_{j}\right)$ as the matching cost of these two points, given by
(5)$$C_{ij}=C\left(p_{i},q_{j}\right)=\frac{1}{2}\sum_{k=1}^{K}\frac{{\left[h_{i}\left[k\right]-h_{j}\left[k\right]\right]}^{2}}{h_{i}\left[k\right]+h_{j}\left[k\right]}$$
where $h_{i}\left[k\right]$ and $h_{j}\left[k\right]$ denote the $k_{th}$ bin histogram at $p_{i}$ and $q_{j}$ respectively.

For all pairs of points $p_{i}$ on the first shape and $q_{j}$ on the second shape, calculate the cost as Equation (Equation 5) and then we got a cost matrix. The next step is to find the optimal alignment between two shapes that minimizes total cost. This can be done by the Hungarian method with time complexity of $O(N^{3})$. The cost between shape contexts is based on the chi-square test statistic that is not a suitable distance metric. Thin plate spline (TPS) model is adopted for modeling transformation [35]. After that, we get the measurable distance of two shapes. The smaller the distance, the more similar these two shapes, or vice versa. So, if the average distance between test signature and reference signatures is lower than a threshold, it would be accepted as a genuine signature of its claimed user.

#### 2.2.2. Trend-Transition-Point Selection

The shape context representation of signature should not only capture specific shape features but also allow considerable variations. Besides, the computational load of distance calculation is closely related to the number of points. Therefore, with the hope of efficiency improvement and variances tolerance, a few representative points are selected. Only selected points can participate in shape distance calculation. In this paper, we propose the TTP selection method.

Trend-transition-points are the points where the curve trends before and after them are completely different while the trends between two successive TTPs do not have obvious change so that the curve shape of the segment approximates to a straight line. So, the signature could be re-constructed with these points. In our method, local extreme points and corner points are all defined as TTPs. The local extreme points are selected depending on its value greater or smaller than its neighborhood. The corner points selection we adopted is proposed in [36,37], which makes use of the smaller eigenvalues of covariance matrices of regions of support.

Let $S_{k}\left(s_{i}\right)$ denotes the region of support (ROS) of point $s_{i}$, a small curve segment containing itself and *k* points in its left and right neighborhoods. That is
$$S_{k}\left(s_{i}\right)=\left[s_{j}=\left(x_{j},y_{j}\right)|j=i-k,i-k+1,\cdots ,i+k-1,i+k\right]$$
where $\left(x_{j},y_{j}\right)$ are the Cartesian coordinates of $s_{i}$.

Therefore, the 2×2 covariance matrix for points in the segment $S_{k}\left(s_{i}\right)$ is calculated. $\lambda_{L}$ and $\lambda_{S}$ are two eigenvalues corresponding to the covariance matrix. The smaller eigenvalues $\lambda_{S}$ can be used to measure prominence of corners over its ROS. In other words, sharper corner points have the large $\lambda_{S}$ and weaker corners have small one. When the points are on a straight line or on a flat curve, the $\lambda_{S}$ will be very small, even approximate to zero. So, corners can be determined if its $\lambda_{S}$ exceeds a predetermined threshold.

Shape contexts are calculated on every point, but only TTPs are used to distance calculation. For every sample point of signatures, the algorithm is implemented as follows.

Step 1:If the point is a start point, add it to TTP dataset. Else, go to Step 2;Step 2:If the point is an end point, add it to TTP dataset and go to Step 5. Else, go to Step 3;Step 3:If the point is an extreme point, add it to TTP dataset. Else, go to Step 4;Step 4:If the point is a corner point, add it to TTP dataset. Else, head to next sample point and return to Step 2;Step 5:For all points in TTP dataset, the point with smaller $\lambda_{S}$ would be deleted when the distance of two successive points is lower than a threshold. The process repeats several times until the distances between points are long enough.

### 2.3. Function Features-Based Online Signature Verification

One of the advantages of online signature verification is that signature is captured by specialized sensors-based devices. So dynamic information can be recorded and used for verification, which makes verification more accurate and reliable. In function features-based methods, a set of function features, such as position, pressure, velocity, acceleration, etc., is firstly captured. Then matching between features of the test and the reference and decision-making are implemented.

#### 2.3.1. Function Features Extraction

Usually, lots of features can be obtained directly from the specialized electronic devices. Horizontal and vertical position, pressure and timestamp of each sample point are the basic measurements. Let x,y,p,t be the mentioned basic measurements, n=1,2,3,…N be the discrete time index of the temporal functions and *N* be the time duration of a signature in sampling units [14]. Based on them, various features can be derived. Among them, 20 frequently used function features are selected. The features are grouped according to their properties, such as position-related, pressure-related, velocity-related, acceleration-related, and angle-related. The features are listed in Table 1.

#### 2.3.2. Matching Based on Shape Context-Dynamic Time Warping (SC-DTW)

Feature matching is very critical for function features-based verification. In recent years, DTW has been widely applied as the matching technique in online signature verification. The DTW method compress or expand the time axis of two temporal functions locally to make them aligned.

Here are two time series $T=(t_{1},t_{2},\ldots ,t_{N})$ and $R=(r_{1},r_{2},\ldots ,r_{M})$ and their lengths are N and M respectively. The similarity between the $n_{th}$ point of *T* and the $m_{th}$ point of *R* are calculated according to defined similarity rule. All the similarity values constitute a DTW cost matrix denoted by d(m,n) defined as:(6)$$d\left(m,n\right)=\|r_{m}-t_{n}\|$$

The overall distance is calculated as following equation:(7)$$D\left(m,n\right)=d\left(m,n\right)+\min\left\{\begin{array}{c} D(m,n-1)+C\\ D(m-1,n-1)\\ D(m-1,n)+C \end{array}\right.$$
where D(n,m) is the cumulative distance up to the current element and C is gap cost. To alleviate the situation of signature at different length, the distance is normalized by Equation (Equation 8).
(8)$$d=\frac{D}{\sqrt{M\times N}}$$

DTW has been an effective method of finding the alignment between two signatures with different length. However, a time series has both numerical nature and shape nature. DTW warps time series depending on the similarity of their numerical characteristics as Equation (Equation 6) but ignores the shape properties. It may lead to abnormal alignment sometimes. Zhang and Tang [38] propose a novel variant of DTW, named SC-DTW. The SC-DTW employs shape context to replace the raw observed values used in conventional DTW, getting ahead in time series data mining. In this paper, we adopted the SC-DTW for function features-based verification to further improve the accuracy.

Specifically, the alignment of two point is decided by their shape matching cost of shape contexts, which means
(9)$$d\left(n,m\right)=C_{nm}$$
where $C_{nm}$ is defined in Equation (Equation 5).

Under this circumstance, a function feature is considered to be a 1-D array and a 2-D shape. The problem of measuring the similarity of two function features can be translate into how similar these two shapes. Figure 4 shows the process of SC-DTW. Figure 4a,b are the time series of $11_{th}$ function feature *v* listed in Table 1 of two signatures from the same user. Figure 4c,d are the corresponding shape context histograms. That the shape context is similar means the sample points in time series are well matched. Please note that the application of shape contexts is only used to find the alignment between two time series. The measurable cumulative distance of them is still obtained by the original cost matrix for the convenience of following classification.

Given a $N^{\left(k\right)}\times D$ feature set $X^{\left(k\right)}$, extracted from a reference signature and a $N^{\left(q\right)}\times D$ feature set $X^{\left(q\right)}$, extracted from a signature which is claimed to belong to the same user, a D-dimensional vector $z^{\left(k,q\right)}$ called ’similarity feature vector’ can be derived by calculating the similarity between each pair of corresponding function features using SC-DTW mentioned above.

#### 2.3.3. Classification Based on Interval-Valued Symbolic Representation

The concept of symbolic data analysis has been applied in the field of document image analysis and cluster analysis. Interval-valued and histogram-valued symbolic representation can represent the variability and distribution of feature values. Guru and Prakash [39] extract global features of signature to form an interval-valued feature vectors and proposed a method for verification and recognition based on the symbolic representation. Pal and Alaei [5] also propose an interval-valued symbolic representation-based method for offline verification. In this paper, we first use the interval-valued symbolic representation to model the similarity features derived from SC-DTW and then build a classifier for verification.

Let $\left[ref_{1},ref_{2},\cdots ,ref_{n}\right]$ be a set of *n* enrolled reference signatures of user. In addition, denote *D* as the similarity feature vector of an user, where $D_{ij}^{r}$ is the SC-DTW distance of feature *r* between signature $ref_{i}$ and $ref_{j}$, as is showed in Table 2. Each user has a feature vector like that. For the $k_{th}$ feature, we compute the statistical mean $\mu_{k}$ and standard deviation $\sigma_{k}$ and the lower and upper bound of interval value can be computed as Equation (Equation 10).
(10)$$\begin{array}{c} sf_{k}=\left(\left[f_{k}^{-},f_{k}^{+}\right],\mu_{k},\sigma_{k}\right)\\ f_{k}^{-}=\mu_{k}-\alpha \sigma_{k}\\ f_{k}^{+}=\mu_{k}+\alpha \sigma_{k}\\ \mu_{k}=mean\left(f_{k}\right)\\ \sigma_{k}=std\left(f_{k}\right) \end{array}$$
where $sf_{k}$ is the symbolic representation of $k_{th}$ feature of a user and includes an interval value and two continuous values. α is a scalar to control the upper and lower limit of each feature. In addition, the symbolic feature vectors are computed for all users and stored in the template base for future verification.

For signature verification problem, the signature is compared with all the reference signatures belonging to the claimed ID. Let $F_{t}=\left[f_{t1},f_{t2},\cdots ,f_{tD}\right]$ denote a D-dimensional feature vector representing the average SC-DTW distance with reference signatures. In addition, denote $sf=[\left[f_{r1}^{-},f_{r1}^{+}\right],\left[f_{r2}^{-},f_{r2}^{+}\right],\cdots ,\left[f_{rD}^{-},f_{rD}^{+}\right]]$ as the reference signatures of the claimed identity described by an interval-valued symbolic feature vector. Each feature value of the test signature is compared with corresponding interval in sf to examine whether it lies within the interval. The feature value represents the dissimilarity of two signatures. That is, the more similar the two signatures, the smaller the value and the closer to 0. The total value of features of a test signature which fall inside the interval value decides how this test signatures is similar to genuine ones, as is showed in Equations (Equation 11) and (Equation 12). Define *A* as the measure of measure of degree of authenticity: (11)$$A=\sum_{i=1}^{D}C\left(f_{ti},\left[f_{ri}^{-},f_{ri}^{+}\right]\right)$$
where
(12)$$C\left(f_{ti},\left[f_{ri}^{-},f_{ri}^{+}\right]\right)=\left\{\begin{array}{cc} 2 & \;\mathrm{if}\;0\leq f_{ti}\leq f_{ri}^{-}\\ 1 & \;\mathrm{if}\;f_{ri}^{-}<f_{ti}\leq f_{ri}^{+}\\ 0 & \;\mathrm{else}\; \end{array}\right.$$

If the acceptance count *A* is greater that a threshold th, the test signature will be classified as a genuine signature of its claimed user. In the user-dependent scenario, every person has its own *A* which is computed using those training samples. For each training signature, there is an *A* we got. For each person, we compute several *A* and then average them thus getting $A_{m}$. The threshold th equals to $\beta \times A_{m}$.

### 2.4. Two-Stage Online Signature Verification

The forgery can be classified to be two types, named skilled forgery and random one. In real applications, random forgeries appear more frequently while skilled forgeries occur less. On the other hand, skilled forgeries are much more difficult to be verified correctly. In this paper, we propose a method using shape contexts and function features as well as a two-stage strategy for accurate online signature verification. The shape context-based verification module is firstly used to reject obvious random forgeries quickly while the function features-based verification module is applied to re-check the signatures survived from the previous module. In this way, the whole system can achieve higher accuracy and consume less computation cost at the mean time.

Two metrics named FRR (False Reject Rate) and FAR (False Accept Rate) have been widely used to evaluate signature verification system. For cascade structure applied in our method, the relationship of FRR and FAR between the sub-verification modules and the whole system are showed in Table 3, where *p* denotes the reject percentage of first sub-verification module. Obviously, *p* takes the value smaller than 1.
(13)$$\begin{array}{cc} r_{1}<r_{2} & \Rightarrow pr_{1}+\left(1-p\right)r_{2}<r_{2}\\ p<1 & \Rightarrow \left(1-p\right)a_{2}<a_{2} \end{array}$$

It can be seen that the performance of the cascade system depends on the thresholds of two sub-verification modules. If p<1 and $r_{1}$ is set to be smaller than $r_{2}$, the cascade system can achieve better performance than the sub-verification modules in terms of false acceptance rate, which is illustrated in Equation (Equation 13).

## 3. Experimental Results

### 3.1. Database and Evaluation Measurement

To evaluate the effectiveness of the proposed method, we run a set of experiments on public database SVC 2004 Task2. There are 40 users and each user has 20 genuine signatures and 20 skilled forgeries. These genuine signatures are collected in two sessions, spaced apart by at least one week. The skilled forgeries are contributed by who could replay the writing sequence of the signatures on the computer screen and practice the forgeries for a few times until they were confident to proceed to the actual data collection. The signatures are mostly in either English or Chinese [40]. In our experiments, for each of the users, we randomly select five male/female genuine signatures for enrolment as reference signatures. The signatures are chosen from the first or second session. The remaining 15 genuine signatures (not selected for enrolment) and 20 skilled forgeries are considered for testing the performance of our proposal. As for the random-forgeries scenario, corresponding to any user, we randomly select 20 signatures from other users. The trial is conducted ten times for each user.

We evaluate the performance primarily using the Equal Error Rate (EER): which is the error when false acceptance rate is equal to false rejection rate. We considered two forms of calculating EER: EER-commonThreshold and EER-userThreshold. EER-commonThreshold is calculated using a global decision threshold. In this case, all the feature values from all training signatures are used to find an optimal value based on minimum EER. The same threshold is shared by all users. EER-userThreshold is using user-specific decision threshold. It is derived from feature values of training samples of each user. For the respective user, the best threshold corresponds to his/her lowest EER. Since there are multiple users in the database SVC 2004, the average of EER across all users is applied as overall performance of the method when using user threshold in our experiments.

### 3.2. Experiment Results

Performance evaluations of shape context (SC)-based verification and function features (FF)-based verification method is firstly conducted. Here Skilled and Random denotes skilled forgeries and random forgeries. In the case of common threshold, the Receiver Operating Characteristic (ROC) curves are given to evaluate the performance. As for user-dependent threshold set-up, EER of every user are expressed as histograms.

The results of these two methods are shown in Figure 5 and Figure 6. From the results, we can see that both the SC and FF method perform well, and the better results are achieved using user thresholds on random forgery verification. It is a general statement that the usage of user threshold usually can yield better performance than common threshold, as is proved by the results. For common threshold, it is difficult to use one value to cover the differences of different individuals. For user threshold, the value is user-specific, varying from one user to another.

As descried in the previous section, 20 frequently used features are categorized into 5 groups according to their properties. To achieve best performance and to investigate contributions of different features, we run a series of experiments. Since only single feature or single feature group cannot provide enough classification ability for online signature verification, we test several combinations of feature groups. For clear illustration, we use G1−G5 to represent the 5 groups: position-related, pressure-related, velocity-related, acceleration-related, and angle-related. The symbol ∪ denotes combination of different groups. The experimental results are given in Table 4. From the results, we can see that using all 20 features performs the best. It is also shown that when velocity-related FF are removed, verification performance deteriorates a lot.

To compare the performances of SC and FF method more clearly, the experimental results of two methods are shown together. Figure 7 gives the results of SC and FF method on skilled forgery while Figure 8 on random one. From the figures, it can be seen that for random forgery verification the performances of SC method and Feature Function method (FF) are similar while FF method outperforms SC method much more on skilled forgery verification. As described in the previous section, SC method is good at extracting global features from signatures with low computation cost, which are quite effective and sufficient for random forgery verification. FF method extracts more detailed features, thus achieving better performance than SC method on skilled forgery verification.

In real applications, random forgeries occur much more frequently that skilled ones. Based on the experimental results, a cascade verification method is designed and tested. The shape context-based verification method is firstly used to reject obvious random forgeries quickly while the function features-based verification method is applied to re-check the signatures survived from the previous module. As illustrated in Section 2.4, FRR of SC method should be smaller than function features-based verification to achieve higher accuracy with lower computation cost. In case of common threshold, FRR of the SC method is set to be 1% and 65% skilled forgeries and 25% random forgeries can be accepted by the second module for re-verification. Figure 9 and Figure 10 give the detailed results on SC method, function feature method, and the two-stage method. Table 5 shows the detailed results on EERs. From the results, it can be seen that the two-stage method achieves the best performance with tolerable computation cost.

Comparisons with the state of the art on database SVC2004 are given in Table 6. It is not easy to make fair comparisons of online signature verification methods due to different databases, training, testing, etc. We select several recently published works which use the same database (SVC2004 ) with us. The method proposed by Lai et al. [28] based on GRU network obtained slightly higher EER than our method. However, it needs more training samples and consumes more computation costs.

## 4. Conclusions

In this paper, we propose a two-stage method using SCs and FF for accurate online signature verification. Features of SCs are extracted from the input firstly and classification of this stage is based on shape distance metric. Only the inputs passing by the first stage are represented by a set of FF and verified. To improve the matching accuracy and efficiency, we propose a SC-DTW to compare the test signature with the enrolled reference ones based on the extracted FF. Then an interval-valued symbolic representation-based classifier is proposed to decide if the test signature is a genuine one. The proposed method is evaluated on SVC2004 Task 2 database achieving an EER of 2.39% which is competitive to the state-of-the-art approaches. The experiment results demonstrate the effectiveness of the proposed method.

## Figures and Tables

**Figure 1 sensors-19-01808-f001:**
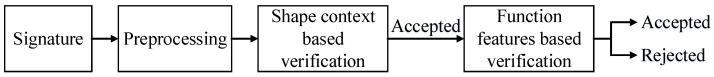
Diagram of proposed verification system.

**Figure 2 sensors-19-01808-f002:**
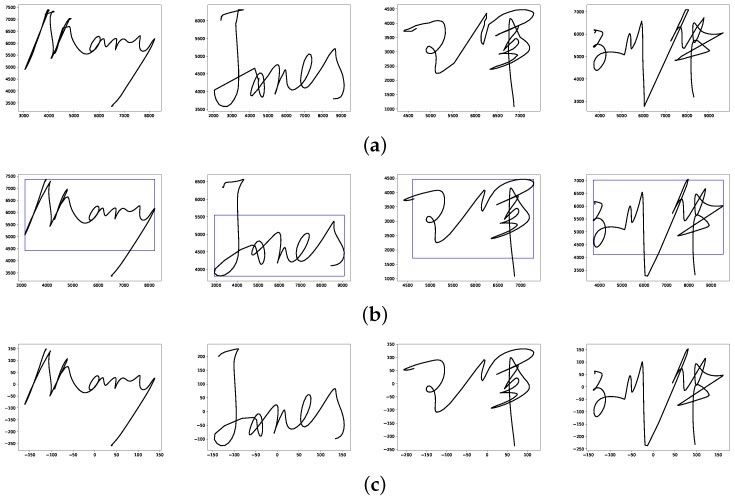
Examples of signature preprocessing. (**a**) Four English or Chinese examples of original signatures. (**b**) Window calculated by moment of signatures. (**c**) Preprocessed results of corresponding signatures.

**Figure 3 sensors-19-01808-f003:**
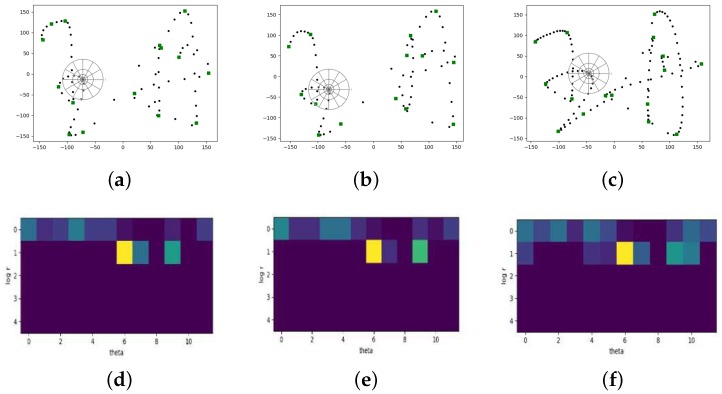
Examples of shape context feature extraction. (**a**) A reference signature of one user. (**b**) A corresponding genuine signature from the same user with reference one. (**c**) A skilled corresponding forgery from the same user with reference one. The green square points represent selected trend-transition-points. (**d**–**f**) Shape context histograms for chosen trend-transition-point in the signatures of (**a**–**c**), respectively.

**Figure 4 sensors-19-01808-f004:**
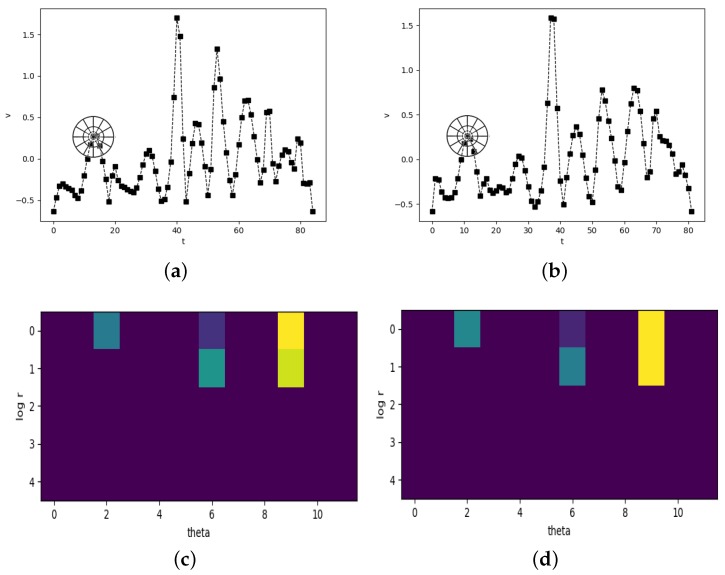
SC-DTW. (**a**,**b**) Time series of total velocity *v* from two signatures and a pair of corresponding points found by shape context. (**c**,**d**) show the shape context histograms of the points marked in (**a**,**b**), respectively.

**Figure 5 sensors-19-01808-f005:**
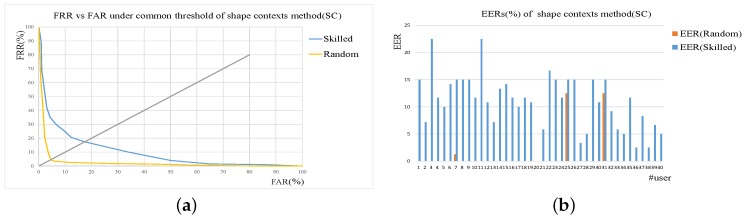
Results of shape context-based verification method (SC). (**a**) ROC curves under common threshold. (**b**) EER of each user under user threshold.

**Figure 6 sensors-19-01808-f006:**
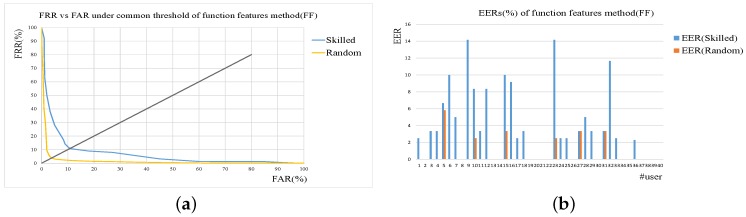
Results of function features-based verification method (FF). (**a**) ROC curves under common threshold. (**b**) EER of each user under user threshold.

**Figure 7 sensors-19-01808-f007:**
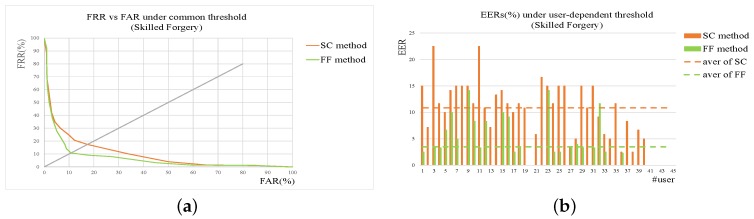
Results of SC and FF method on skilled forgery. (**a**) ROC under common threshold. (**b**) EER of each user under user threshold. In addition, those dotted lines are the average levels of corresponding methods.

**Figure 8 sensors-19-01808-f008:**
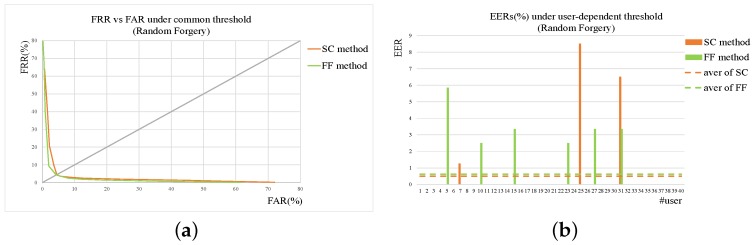
Results of SC and FF method on random forgery. (**a**) ROC under common threshold. (**b**) EER of each user under user threshold. In addition, those dotted lines are the average levels of corresponding methods.

**Figure 9 sensors-19-01808-f009:**
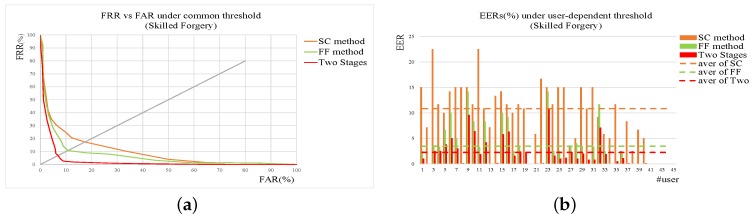
Results of SC, FF, and two-stage method on skilled forgery. (**a**) ROC under common threshold. (**b**) EER of each user under user threshold. In addition, those dotted lines are the average levels of corresponding methods.

**Figure 10 sensors-19-01808-f010:**
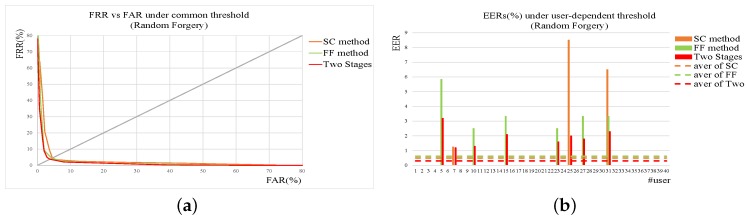
Results of SC, FF, and two-stage method on random forgery. (**a**) ROC under common threshold. (**b**) EER of each user under user threshold. In addition, those dotted lines are the average levels of corresponding methods.

**Table 1 sensors-19-01808-t001:** Function features extracted for online signature verification.

Category	Description	Symbols
Position-related	*x* coordinate	x(n)
	*y* coordinate	y(n)
	Displacement	$S\left(n\right)=\sqrt{x{\left(n\right)}^{2}+y{\left(n\right)}^{2}}$
	Change of *x* coordinate	$\Delta x_{n}=x\left(n+1\right)-x\left(n\right)$
	Change of *y* coordinate	$\Delta y_{n}=y\left(n+1\right)-y\left(n\right)$
	Change of displacement	$\Delta S\left(n\right)=\sqrt{{\left(\Delta x\left(n\right)\right)}^{2}+{\left(\Delta y\left(n\right)\right)}^{2}}$
Pressure-related	Pressure	p(n)
	Change of pressure	$\Delta p_{n}=p\left(n+1\right)-p\left(n\right)$
Velocity-related	*x* velocity	$v_{x}\left[n\right]=\frac{x(n+1)-x(n)}{t(n+1)-t(n)}$
	*y* velocity	$v_{y}\left[n\right]=\frac{y(n+1)-y(n)}{t(n+1)-t(n)}$
	Total velocity	$v\left(n\right)=\sqrt{v_{x}^{2}\left(n\right)+v_{y}^{2}\left(n\right)}$
Acceleration-related	*x* acceleration	$a_{x}\left[n\right]=\frac{v_{x}\left(n+1\right)-v_{x}\left(n\right)}{t(n+1)-t(n-1)}$
	*y* acceleration	$a_{y}\left[n\right]=\frac{v_{y}\left(n+1\right)-v_{y}\left(n\right)}{t(n+1)-t(n-1)}$
	Total acceleration	$a\left(n\right)=\sqrt{a_{x}^{2}\left(n\right)+a_{y}^{2}\left(n\right)}$
	Centripetal acceleration	$a_{c}\left(n\right)=\left[v_{x}\left(n\right)\cdot a_{y}\left(n\right)-v_{y}\left(n\right)\cdot a_{x}\left(n\right)\right]/v\left(n\right)$
Angle-related	Cosine of the angle between *x*-axis and signature curve	$\cos\alpha =\frac{x(n+1)-x(n)}{\sqrt{{\left(x\left(n+1\right)-x\left(n\right)\right)}^{2}+{\left(y\left(n+1\right)-y\left(n\right)\right)}^{2}}}$
	Sine of the angle between *x*-axis and signature curve	$\sin\alpha =\frac{y(n+1)-y(n)}{\sqrt{{\left(x\left(n+1\right)-x\left(n\right)\right)}^{2}+{\left(y\left(n+1\right)-y\left(n\right)\right)}^{2}}}$
	Cosine of the angle between *x* velocity and total velocity	$\cos\beta =v_{x}\left(n\right)/v\left(n\right)$
	Angle between *x*-axis and signature curve	$\theta \left(n\right)=\tan^{-1}\frac{y(n+1)-y(n)}{x(n+1)-x(n)}$
	Angle velocity	$v_{\theta }\left(n\right)=\frac{\theta (n+1)-\theta (n)}{t(n+1)-t(n)}$

**Table 2 sensors-19-01808-t002:** Similarity feature vector of each individual.

	Fea.	$f_{1}$	$f_{2}$	⋯	$f_{r}$	⋯	$f_{D}$
Ref.							
$ref_{1}/ref_{2}$		$D_{12}^{1}$	$D_{12}^{2}$	⋯	$D_{12}^{r}$	⋯	$D_{12}^{D}$
$ref_{1}/ref_{3}$		$D_{13}^{1}$	$D_{13}^{2}$	⋯	$D_{13}^{r}$	⋯	$D_{13}^{D}$
⋯		⋯	⋯	⋯	⋯	⋯	⋯
$ref_{2}/ref_{3}$		$D_{23}^{1}$	$D_{23}^{2}$	⋯	$D_{23}^{r}$	⋯	$D_{23}^{D}$
$ref_{2}/ref_{4}$		$D_{24}^{1}$	$D_{24}^{2}$	⋯	$D_{24}^{r}$	⋯	$D_{24}^{D}$
⋯		⋯	⋯	⋯	⋯	⋯	⋯
$ref_{i}/ref_{j}$		$D_{ij}^{1}$	$D_{ij}^{2}$	⋯	$D_{ij}^{2}$	⋯	$D_{ij}^{D}$
⋯		⋯	⋯	⋯	⋯	⋯	⋯

**Table 3 sensors-19-01808-t003:** FRR and FAR of individual verification modules and cascade system.

System Framework	FRR	FAR
Shape Context Module	$r_{1}$	$a_{1}$
Feature Function Module	$r_{2}$	$a_{2}$
Cascade of two Modules	$pr_{1}+\left(1-p\right)r_{2}$	$\left(1-p\right)a_{2}$

**Table 4 sensors-19-01808-t004:** Comparisons between different group of function features.

Group	Not Included Feature Group	EER(SF)	EER(RF)
G1∪G2∪G3∪G4∪G5	None	10.8	4.5
G1∪G2∪G3∪G4	Angle-related	11.5	4.8
G1∪G2∪G3∪G5	Acceleration-related	12.2	6.3
G1∪G2∪G4∪G5	Velocity-related	13.5	7.2
G1∪G3∪G4∪G5	Pressure-related	11.8	6.8
G2∪G3∪G4∪G5	Position-related	11.2	5.9

**Table 5 sensors-19-01808-t005:** Verification results (EER%) of different methods with common threshold and user threshold.

Method	Time(s)	Common Threshold		User Threshold	
		EER(SF)	EER(RF)	EER(SF)	EER(RF)
Shape context-based verification	0.47	17.4	4.5	10.45	0.5
Function features-based verification	1.26	10.8	4.3	3.5	0.62
Two-stage verification	1.04	6.92	3.8	2.39	0.3

**Table 6 sensors-19-01808-t006:** Comparisons with the state-of-the-art works on database SVC2004.

Works	Method	EER(%)
Song et al., 2016, [41]	DTW with SCC	2.89
Liu et al., 2017, [42]	Spare representation	2.98
Xia et al., 2018, [6]	GMM+DTW with SCC	2.63
Sharma et al., 2018, [9]	DTW+warping path alignment	2.53
Lai et.al., 2017, [28]	LNPS+GRU	2.37
Proposed method	Two-stage verification	2.39

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
