# Peer review of "A Two-Stage Method for Online Signature Verification Using Shape Contexts and Function Features"

_sensors, 2019, doi:10.3390/s19081808_

Round 1

Reviewer 1 Report

My comments are as follows: 1. Overall the quality of the manuscript is good. It could be accepted with minor modification. 2. Authors are requested to mention their contributions at the end of the introduction section in bullet form. 3. Authors should mention how the submitted manuscript differs from "Jia, Y.; Huang, L. Online Signature Verification Based on Shape Context and Function Features. In 372 Proceedings of Chinese Conference on Pattern Recognition and Computer Vision, 2018, pp. 62–73." 4. Please cite the following paper, where face biometric is mentioned (second line of the introduction section). Ayan Seal, Debotosh Bhattacharjee, Mita Nasipuri, Consuelo Gonzalo-Martin, Ernestina Menasalvas, “A-trous Wavelet Transform based Hybrid Image Fusion for Face Recognition using Region Classifiers”, Expert Systems, Wiley, doi.org/10.1111/exsy.12307, 2018.

Author Response

Thank you for the comments and suggestions.
Point 1: Authors are requested to mention their contributions at the end of the introduction section in bullet form.
Response 1: According to the suggestion, contributions are given in bullet form at the Section “Introduction”.
Point 2: Authors should mention how the submitted manuscript differs from "Jia, Y.; Huang, L. Online Signature Verification Based on Shape Context and Function Features. In 372 Proceedings of Chinese Conference on Pattern Recognition and Computer Vision, 2018, pp. 62–73."
Response 2: The illustration on the differences from the one published in proceedings of PRCV2018 is given at the second to the last paragraph of Section “Introduction”.
Point 3: Please cite the following paper, where face biometric is mentioned (second line of the introduction section).
Ayan Seal, Debotosh Bhattacharjee, Mita Nasipuri, Consuelo Gonzalo-Martin, Ernestina Menasalvas, “A-trous Wavelet Transform based Hybrid Image Fusion for Face Recognition using Region Classifiers”, Expert Systems, Wiley, doi.org/10.1111/exsy.12307, 2018.
Response 3: Face recognition is one of the biometric verification as introduced in the manuscript, so the paper on face recognition mentioned has been cited.

Reviewer 2 Report

The authors proposed an interesting approach for verifying online signature by means of an algorithm which processes shape context descriptors as well as such hand crafted features.

The comparison approach between processed signature with reference is performed by means of a Dynamic Time Warping methodology

The method seems promising.

The authors are suggested to extend the prior art comparison as they missed to report a robust comparison with such approaches based on usage of CNNs or LSTMs/Autoeconder. In fact, one of the most current scientific open discussion is the comparison between the features extracted from recent machine learning systems against to ad-hoc defined hand crafted features: add this kind of comparison as of course it is needed to improve the robustness of the proposed results.

The authors in table 1 reported the designed hand crafted features but it would be worthwhile to better explain the meaning of each feature and the single impact on the overall performance of the proposed method. 

Moreover, the authors are suggested to compare their used features with similar ones proposed in literature and usually used when an image or signal has to be mathematically described, as the features proposed in the following paper:

Conoci, S., Rundo, F., Petralia, S., & Battiato, S. (2017). Advanced skin lesion discrimination pipeline for early melanoma cancer diagnosis towards PoC devices. 2017 IEEE Proceedings of European Conference on Circuit Theory and Design (ECCTD), 1-4.

Finally, the authors are suggested to better explain the introduced (section 3.2)  method based on common threshold / user threshold with regard to EER % results, as that explanation  is not very clear.

Author Response

Thank you for the comments and suggestions.
Point 1: The authors are suggested to extend the prior art comparison as they missed to report a robust comparison with such approaches based on usage of CNNs or LSTMs/Autoeconder. In fact, one of the most current scientific open discussions the comparison between the features extracted from recent machine learning systems against to ad-hoc defined hand crafted features: add this kind of comparison as of course it is needed to improve the robustness of the proposed results.
Response 1: Yes, as you said, with the great success achieved by CNNs or LSTMs on image classification or sequential modelling tasks, there are several works of applying deep learning based methods to signature verifications. So, as you suggested, we supplement such contents in sixth paragraph of Section “Introduction”. And, we also add the comparison between the proposed method and the deep learning based method, which is given in Table 6 of Section 3.2.
Point 2: The authors in table 1 reported the designed hand crafted features but it would be worthwhile to better explain the meaning of each feature and the single impact on the overall performance of the proposed method.
Response 2: The features listed in Table 1 actually can be categorized into several groups, so as suggested, we revised the illustration method of features in Table 1 to make them clearer. The impact of feature groups is investigated and the results are given in Table 4. Besides, since only single feature or single feature group cannot provide enough classification ability for online signature verification problem, we test several combinations of feature groups and try to find the meaningful conclusion of impact on verification performances. The discussions are given in corresponding to Table 4.
Point 3: Moreover, the authors are suggested to compare their used features with similar ones proposed in literature and usually used when an image or signal has to be mathematically described, as the features proposed in the following paper:
Conoci, S., Rundo, F., Petralia, S., & Battiato, S. (2017). Advanced skin lesion discrimination pipeline for early melanoma cancer diagnosis towards PoC devices.2017 IEEE Proceedings of European Conference on Circuit Theory and Design (ECCTD), 1-4.
Response 3: We read the above mentioned paper and found that the features applied in the paper are designed for static images while in our problem, on-line signature verification, the inputs are dynamic data. Therefore, the feature extraction method of the paper can hardly applied to our problem.
Point4: Finally, the authors are suggested to better explain the introduced (section 3.2) method based on common threshold / user threshold with regard to EER % results, as that explanation is not very clear.
Response 4: Yes, as suggested, more detailed descriptions about how to set up the environment of common threshold and user threshold are added in Section 3.1 and more detailed explanations of the experiment results are given in the Section 3.2.

Round 2

Reviewer 2 Report

The author revised the paper as suggested. It can be accepted as it is.